

# Enhancing Accuracy of Indoor Air Quality Sensors via Automated Machine Learning Calibration

Juncheng Qian[1], Thomas Wynn[1], Bowen Liu[2], Yuli Shan[1], Suzanne E. Bartington[3], Francis D. Pope[1], Yuqing Dai[1,*], Zongbo Shi[1,*]

5    [1] School of Geography, Earth and Environment Sciences, University of Birmingham, Birmingham, B15 2TT, UK
[2] Department of Management, Birmingham Business School, University of Birmingham, Birmingham, B15 2TT, UK
[3] Institute of Applied Health Research, University of Birmingham, Birmingham, UK

*Correspondence to*: Zongbo Shi (z.shi@bham.ac.uk), Yuqing Dai (y.dai.2@bham.ac.uk)

25

30



**Abstract.** Indoor fine particles (PM$_{2.5}$) exposure poses significant public health risks, prompting growing use of low-cost sensors for indoor air quality monitoring. However, maintaining data accuracy from these sensors is challenging, due to interference of environmental conditions, such as humidity, and instrument drift. Calibration is essential to ensure the accuracy of these sensors. This study introduces a novel automated machine learning (AutoML)-based calibration framework to enhance the reliability of low-cost indoor PM$_{2.5}$ measurements. The multi-stage calibration framework connects low-cost field sensors to be deployed with intermediate drift-correction reference sensors and a reference-grade instrument, applying separate calibration models for low (clean air environment) and high (pollution events) concentration ranges. We evaluated the framework in a controlled indoor chamber using two different sensor models exposed to diverse indoor pollution sources under uncontrolled natural ambient conditions. The AutoML-driven calibration significantly improved sensor performance, achieving a strong correlation with reference measurements (R$^2$>0.90) and substantially reducing error metrics (with root-mean-square error (RMSE) and mean absolute error (MAE) roughly halved relative to uncalibrated data). Bias was effectively minimised, yielding calibrated readings closely aligned with the reference instrument. These findings demonstrate that our calibration strategy can convert low-cost sensors into a more reliable tool for indoor air pollution monitoring. The improved data quality supports atmospheric science research by enabling more accurate indoor PM$_{2.5}$ monitoring, and informs public health interventions and evaluation by facilitating better indoor exposure assessment.

## 1 Introduction

Air quality monitoring is essential in understanding exposure to pollutants in both outdoor and indoor environments, which informs public health improvement strategies. In particular, indoor air quality (IAQ) has gained attention because people spend the majority of their time indoors, yet historically it has been difficult to measure indoor pollutants continuously (Aix et al., 2023). Traditional approaches for IAQ assessment relied on expensive reference instruments (e.g. filter-based gravimetric samplers with pumps and impactors) that require expert operation and maintenance. These practical challenges made long-term indoor monitoring infeasible in most settings (Levy Zamora et al., 2018). Recently, however, dramatic advances in low-cost sensor technology have transformed this landscape. Compact and affordable low-cost sensors for particulate matter (PM) and gases have made it possible to deploy dense monitoring networks and to track air quality in homes, offices, and other indoor spaces in real-time. For example, a consumer-grade PM sensor "PurpleAir" is now widely used, and over 5,600 devices reporting to an online map, and about 18% of these were deployed indoors as of 2020 (Koehler et al., 2023). This surge in low-cost sensor use highlights their promise for broad IAQ surveillance and community engagement in air quality improvement efforts.

As low-cost sensors proliferate, ensuring their data quality through proper calibration has become a critical concern. These sensors often suffer from biases and interferences that can compromise accuracy. For example, low-cost PM sensors that use optical scattering can be highly sensitive to environmental factors like relative humidity (RH) and aerosol properties. At high RH (> 80%), condensation on the sensor or particles can lead to overestimation of fine particles (PM$_{2.5}$) concentrations (Crilley





et al., 2020; Hagan & Kroll, 2020). Cross-sensitivities are also common, electrochemical gas sensors may respond to non-target gases (e.g. ozone sensors responding to nitrogen dioxide $NO_2$). Moreover, the performance of air quality sensors can degrade over time due to aging and fouling of components (so-called "drift effect"). Studies have showed that low-cost sensors tend to lose sensitivity or shift baseline after months of use, and electrochemical sensor singles degrades within two years, necessitating periodic recalibration (Zaidan et al., 2022; Zimmerman et al., 2018) .


To address these issues, a variety of calibration techniques have been explored previously, ranging from simple corrections to machine learning (ML) models. Traditional calibration methods typically include collocating low-cost sensors with a reference-grade instrument (such as federal reference methods, FRMs) and deriving a statistical correction (Liang, 2021). The simplest approach is a linear regression or affine transformation that aligns the sensor readings to the reference values.

Additional environmental parameters are generally incorporated into multi-variate calibration models, for example, temperature and RH are included as independent variables to account for their influence on sensor response (Kang & Choi, 2024). These methods, including one-point or two-point calibrations and polynomial fits, have been shown to improve sensor accuracy under stable conditions (Cowell et al., 2023). In practice, laboratories or field researchers may perform a pre-deployment calibration by exposing sensors to known pollutant concentrations and fitting a curve. However, a calibration

derived in one setting does not necessarily transfer well to another. Studies have noted that calibrations done in controlled lab environments often do not span the full range of real-world conditions, limiting their generality (Kim et al., 2019; Li et al., 2018; Mousavi & Wu, 2021). Different particle compositions also affect the magnitude of the sensor response (Crilley et al., 2020; Zou et al., 2021). Therefore, in situ calibration is often recommended to capture local environmental effects to yield more robust calibration models, allowing necessary adjustments for factors like aerosol composition and meteorological

conditions (Raysoni et al., 2023). Although the performance of these traditional methods may be suboptimal when sensor response relationships are highly non-linear or environment-specific, they are still widely used due to their transparency and ease of implementation.

Recently, ML algorithms have been employed to improve calibration accuracy and capture complex sensor behaviours. ML calibration methods can simulate non-linear relationships and interactions that traditional linear methods might neglect

(Villanueva et al., 2023). A range of ML approaches has been applied, including artificial neural networks (ANN), support vector regression (SVR), random forests (RF), gaussian process regression (GPR), and even semi-parametric models like generalized additive models (GAM) (Mahajan & Kumar, 2020). These data-driven models leverage not only raw readings from the sensor but often additional features (e.g., RH, temperature, timestamps) to learn the mapping to actual pollutant concentrations. Several studies have presented the effectiveness of ML-based calibration. Nowack et al. (2021) compared a

regularized linear model (ridge regression) against non-linear models (random forest and GPR) for calibrating nitrogen dioxide ($NO_2$) and particulate matter with a diameter less than 10 micrometres ($PM_{10}$) sensors, finding that the machine learning approaches achieved high out-of-sample accuracy (frequently coefficient of determination $R^2>0.8$) and outperformed traditional multiple linear regression models (Nowack et al., 2021). Mahajan et al. (2019) observed that an SVR model



provided better calibration performance for $PM_{10}$ sensors than both linear regression and standard neural networks (Munir et
al., 2019). Nonetheless, ML-based calibrations also present challenges. They typically require a substantial dataset of sensor
as well as reference readings for training, and their predictions can be unreliable outside the range of training data. For instance,
an ANN or RF may struggle to extrapolate to pollutant levels higher than it has been seen during calibration, whereas a
Gaussian process regression model may handle extrapolation with less bias (Nowack et al., 2021). Additionally, the calibration
model learned at one location may not generalize to a new location (i.e., site transferability issue) unless a wide variety of
conditions are considered. Despite these limitations, ML-based calibration can significantly improve the performance of low-
cost sensors when carefully applied (Liu et al., 2019; Nowack et al., 2021; Villanueva et al., 2023; Zimmerman et al., 2018).
While most field calibration studies to date have focused on outdoor deployments, where sensors are co-located with
regulatory-grade monitors or used in ambient networks, a critical gap in the current literature is the calibration of low-cost
sensors specifically for indoor environment.

Indoor air, however, can differ markedly from outdoor air in composition and dynamics. Factors like indoor-generated particles
(from cooking, smoking, etc.), confined space, and higher humidity or temperature fluctuations can all influence sensor
readings. For example, cooking can release ultrafine particles and organic aerosols in short bursts, causing sharp concentration
spikes. A study reported that indoor $PM_{2.5}$ levels peaking near 488 µg m$^{-3}$ during cooking in a home, far exceeding typical
outdoor concentrations (Cowell et al., 2023). Tobacco smoke similarly produces dense particulate matter and complex
chemicals in confined spaces. Also, indoor spaces often have limited ventilation, allowing pollutants to accumulate and
humidity to fluctuate in ways not seen outdoors. These conditions test the limits of calibration models. A calibration model
trained mostly on moderate outdoor pollution levels may not extrapolate well to the abrupt spikes or ultra-low concentrations
encountered indoors (Koehler et al., 2023). Compounding the issue, gathering extensive indoor calibration datasets is difficult,
reference-grade indoor measurements are rare because deploying instruments indoors at scale is resource-intensive. As a result,
there is a paucity of calibration methods tailored to indoor use, and questions remain about how well the algorithms proven in
ambient air translate to indoor settings. This gap is increasingly problematic as the adoption of indoor air quality sensors grows;
without reliable calibration, the data from these sensors could mislead users or undermine trust in sensor-based monitoring.

In this study, we aim to bridge the gap by introducing a replicable calibration approach for indoor air quality sensors using
Automated Machine Learning (AutoML). AutoML is an emerging technology that automates the selection of machine learning
algorithms and hyperparameters to build optimal models (LeDell & Poirier, 2020). Our objective is to develop a calibration
framework that can be easily applied to low-cost sensor data in indoor environment to improve its accuracy and reliability.
Unlike traditional calibration methods that might rely on fixed formulas or manually crafted ML models, an AutoML-based
approach automates the selection and optimization of the calibration model. In our framework, sensor readings (e.g., raw $PM_{2.5}$
concentrations) are combined with environmental variables (mainly indoor temperature and RH), and an AutoML is employed
to identify the best-performing calibration model through automated testing of many algorithms and hyperparameter settings.
By allowing the AutoML system to explore a wide range of potential models (from linear regressions to complex ensemble





methods), we ensure that the final chosen model is well-suited to the characteristics of the indoor dataset, without requiring the user to have advanced machine learning expertise. The proposed approach is replicable in that it provides a general template

that can be applied to other indoor sensor deployments, that is, researchers or practitioners can feed their co-location data into the same AutoML pipeline to obtain a custom calibration model for their specific environment.

The remainder of this paper is structured as follows. Section 2 describes the experimental setup and calibration methodology, including indoor air quality sensors, reference instruments, data collection procedures, and the AutoML workflow employed to generate calibration models. Section 3 presents the calibration results and discusses the implications of the findings. Section

4 summarizes the key findings. We also discuss limitations of our approach and provide recommendations for future research.

## 2 Method

### 2.1 Experimental Configurations

A controlled laboratory experiment was conducted within a custom-built container designed to simulate realistic indoor air pollution conditions (Fig. 1(a)). The chamber was equipped with fans to ensure uniform pollutant distribution (Fig. 1(b)),

which minimized spatial concentration variations, essential for maintaining stable and reproducible conditions during sensor evaluation. An aerosol spectrometer (i.e., Palas Fidas 200) was employed as the reference-grade instrument to provide high-precision baseline measurements for sensor performance evaluation and calibration. A total of 40 low-cost air quality sensors was deployed within the chamber, settled on a table at near the same height with Fidas 200 to minimize positional variability. Our air quality sensors consisted of two different types, including 20 units of AirGradient ONE (Model I-9PSL) and 20 units

of AtmoCube. AirGradientONE sensors measure $PM_{2.5}$ using a Plantower PMS5003 laser-scattering sensor, and temperature and RH through a Sensirion SHT40 sensor. AtmoCube sensors detect particulate matter using a Sensirion SPS30 laser-scattering sensor, temperature using a Sensirion STS35-DIS, and RH using a Sensirion SHTC3.

To generate diverse and realistic indoor air pollution profiles, three indoor emission sources were introduced into the experimental container, including incense sticks, cigarette smoke from 7[th] to 21[st] Oct 2024, and cooking emissions (i.e., frying

vegetables, bacon, and fries) from 22[nd] to 30[th] Oct 2024 (Fig. 1(b) and Fig. 1(c)). All AirGradientONE and AtmoCube sensors and the Fidas 200 were exposed to the same emission sources simultaneously. Temperature and RH levels were allowed to exchange passively with the ambient environment with no ventilations or windows opening, mimicking natural indoor conditions where these parameters fluctuate freely. Between each emission event, the container was ventilated until pollutant concentrations returned to background levels (mainly during the night), ensuring that there was no cross-contamination

between different test conditions, thus generating a reliable dataset for subsequent sensor performance evaluation and calibration.





**Figure 1: Overview of indoor air quality sensor calibration setup: (a) fully renovated half-size container, (b) emission sources and analytical instrumentation, and (c) schematic of pollutant generation and instrument placement.**

## 2.2 Automated Machine Learning

We employed an AutoML framework to develop and select calibration models for the indoor air quality sensors. The AutoML approach systematically generate a variety of (i.e., 30 in this study) candidate models and optimises their hyperparameters. In our implementation, the input features to each model included the sensor's raw readings, indoor temperature, and RH, while the target output was the PM concentration measured by Fidas 200. The AutoML process explored multiple regression



algorithms, including gradient boosting machines (GBM), distributed random forest (DRF), and extreme gradient boosting (xgboost), to identify a model that best maps the sensor outputs to the reference concentrations.

A typical training strategy was applied, with 80% of dataset allocated for model training and the remaining 20% reserved for
performance testing.

Evaluation metrics were calculated for each candidate to guide the selection of the best model. We primarily used the root mean square error (RMSE), mean absolute error (MAE), mean bias error (MBE), index of agreement (IOA), and $R^2$ as the performance criteria. RMSE quantified the average magnitude of prediction errors in units matching the observed data, with lower values reflecting smaller deviations. MAE measured the average absolute difference between observed and predicted
values, providing an interpretable measure of accuracy independent of error direction. MBE provides the average bias in the predictions, where positive or negative values indicated overestimation or underestimation, respectively. IOA indicates the overall level of agreement (from -1 to 1) between reference measurements and predicted values, with 1 denoting perfect agreement (ideal model performance), 0 with no agreement (predictions no better than simply predicting the observed average), and -1 with complete disagreement or systematic inverse relationship (Willmott et al., 2011). $R^2$ (values in [0, 1]) indicates the
proportion of the variance in the reference measurements explained by the model, with values closer to 1 indicating a stronger linear association. The formulas are represented below:

$$RMSE = \sqrt{\frac{1}{n}\sum_{i=1}^{n}(o_i - p_i)^2} \tag{1}$$

$$MAE = \frac{1}{n}\sum_{i=1}^{n}|o_i - p_i| \tag{2}$$

$$MBE = \frac{1}{n}\sum_{i=1}^{n}(o_i - p_i) \tag{3}$$

$$IOA = \begin{cases} 1 - \frac{\sum_{i=1}^{n}|p_i - o_i|}{c\sum_{i=1}^{n}|o_i - \bar{o}|}, & when \ \sum_{i=1}^{n}|p_i - o_i| \leq c\sum_{i=1}^{n}|o_i - \bar{o}| \\ \frac{c\sum_{i=1}^{n}|o_i - \bar{o}|}{\sum_{i=1}^{n}|p_i - o_i|} - 1, & when \ \sum_{i=1}^{n}|p_i - o_i| > c\sum_{i=1}^{n}|o_i - \bar{o}| \end{cases} \tag{4}$$

$$R^2 = 1 - \frac{\sum_{i=1}^{n}(o_i - p_i)^2}{\sqrt{\sum_{i=1}^{n}(o_i - \bar{o})^2}} \tag{5}$$

here $o_i$ denotes the $i$-th value from the reference dataset, $p_i$ is the $i$-th predicted value from the calibration models, $n$ represents the total number of data points in the dataset, and $\bar{o}$ is the arithmetic average of all reference measurements. Among all candidate models, stacked ensemble models show superior stability and predictive accuracy and was therefore selected as the
final calibration model in this study (Table S1).



## 2.3 Calibration Procedure

To ensure reproducible calibration of the low-cost sensors against the Fidas 200, we first established a three-step protocol that accounts for variability among sensor units while maintaining consistency with reference measurements. The approach is designed to be scalable for large sensor networks in real-world indoor monitoring applications. The key steps include:

(1) **Field sensor-to-"Drift–reference sensor" calibration (f2d).** A subset of five sensors from each sensor type (AtmoCube and AirGradientONE) was randomly selected to serve as "drift–reference sensors". These drift–reference sensors were used exclusively for calibration purposes and were not deployed for field indoor monitoring. The remaining sensors, referred to as "field sensors", were intended for operational deployment. We employed AutoML to develop calibration models that map the field sensors' raw readings to the corresponding averaged measurements

of the drift–reference sensors at each time step:

$$\widehat{d_j}(t) = \mathcal{F}_j^{f2d}\left(x_j(t)\right) \tag{6}$$

$$\mathcal{F}_j^{f2d} = \arg\min_{f \in \mathcal{F}} \sum_{t=1}^{N}\left[f\left(x_j(t)\right) - \bar{d}(t)\right]^2 \tag{7}$$

$$x_j(t) = \left[s_j(t), T_j(t), RH_j(t)\right]^T \tag{8}$$

$$\bar{d}(t) = \sum_{k=1}^{K} d_k(t) \tag{9}$$

where $\widehat{d_j}(t)$ is calibrated PM concentration for field sensor $j$ (1, …, $M$) at a time index of calibration record $t$ (1, …, $N$); $x_j(t)$ represents raw sensor reading, temperature, and RH; $\bar{d}(t)$ denotes mean of $K$(=5) drift–reference sensors; and $\mathcal{F}_j^{f2d}$ represent best-performing model chosen for sensor $j$ (GBM in this study) from pool of AutoML candidate models $\mathcal{F}$ during this f2d process. Note that here $T_j(t)$ and $RH_j(t)$ should be calibrated against averaged values of the drift–reference sensors using a simple univariate transfer function before being used as input features.

(2) **"Drift–reference sensor" to "Reference instrument" calibration (d2r).** The averaged readings from drift–reference sensors were calibrated against Fidas 200 following similar procedure above:

$$\hat{r}(t) = \mathcal{F}^{d2r}\left(z(t)\right) \tag{10}$$

$$\mathcal{F}^{d2r} = \arg\min_{f \in \mathcal{F}} \sum_{t=1}^{N}\left[f\left(z(t)\right) - r(t)\right]^2 \tag{11}$$

$$z(t) = \left[\bar{d}(t), \bar{T}(t), \overline{RH}(t)\right]^T \tag{12}$$

here $\hat{r}(t)$ represents calibrated PM concentration for drift–reference sensors; $r(t)$ is PM concentration measured by the reference instruments (Fidas 200); $z(t)$ represents a vector of $\bar{d}(t)$, and calibrated $\bar{T}(t)$ and $\overline{RH}(t)$ (against Fidas 200); and $\mathcal{F}^{d2r}$ denotes best-performing model for the d2r calibration.

     Our exploratory analysis (Fig. S1) revealed a clear threshold at 50 µg m$^{-3}$ where the sensor bias flips. The sensors tend to overestimate Fidas 200 measurements below but underestimate them above the threshold. Therefore, we

applied stratified calibration strategy, training separate AutoML models for the low (<50 µg m$^{-3}$) and high (50–600



µg m⁻³) regimes in both the field-to-drift (f2d) and drift-to-reference (d2r) stages. It allows us to tailor the calibration to the specific bias profile of each regime and thereby minimises systematic error across the sensor's full operating range.

**(3) Field sensor-to-"Reference instrument" calibration (f2r).** For every time stamp $t$, the field sensor's raw reading is first converted to a drift–reference proxy as in Step (1) f2d. That proxy, combined with calibrated temperature and RH (against Fidas 200), is then fed into the calibration models in Step (2) d2r to calculate concentrations directly comparable to the reference dataset:

$$\widetilde{r_j}(t) = \mathcal{H}_j\left(x_j(t)\right) \equiv \left(\mathcal{F}^{d2r} \circ \mathcal{F}_j^{f2d}\right)\left(x_j(t)\right) \tag{13}$$

where $\widetilde{r_j}(t)$ denotes final PM concentration of sensor $j$ aligned to Fidas 200; and $\mathcal{H}_j$ represents shorthand for the overall transfer function $\mathcal{F}^{d2r} \circ \mathcal{F}_j^{f2d}$.

The sensor performance drift over long deployments, the calibration derived pre-deployment gradually becomes less reliable. After retrieval we therefore rebuild the f2d and d2r models with the post-deployment dataset, obtaining a second set of predictions $\widetilde{r_j'}(t)$. For any timestamp $t$ within the deployment period $0 \leq t \leq D$ (with $D$ the total duration), we fuse the two predictions with a simple linear weight that shifts emphasis from the pre- to the post-deployment model:

$$r_j * (t) = \left(1 - \frac{t}{D}\right) \times \widetilde{r_j}(t) + \frac{t}{D} \times \widetilde{r_j'}(t) \tag{14}$$

thus, $r_j * (t)$ equals the pre-deployment estimate at the campaign start ($t = 0$), the post-deployment estimate at the end ($t = D$), and a smoothly blended value in between, providing a first order correction for drift. The overall calibration framework is shown schematically in Fig. 2.

## 3 Results and Discussions

### 3.1 Low-cost sensor raw readings

Figure 3 compares the timeseries responses of the two sensor types, from AirGradientONE and AtmoCube to indoor emission events. During the combustion episodes (cigarette smoking and incense-burning) that occurred between 12th and 22nd October 2024, the AirGradientONE sensors repeatedly recorded uncalibrated PM$_{2.5}$ concentrations exceeding 500 µg m⁻³, and all units tracked those peaks almost identically, showing high intra-sensor coherence and a high sensitivity to combustion-derived particles. The AtmoCube sensors followed the same temporal pattern but with systematically lower maximum concentrations compared to the AirGradientONE sensors, with peak readings between 400-to-500 µg m⁻³. Cooking activities generated far lower PM concentrations. Routine meal preparation produced brief excursions of ~30 µg m⁻³ on both sensor types, while a single spike of 80 µg m⁻³ on 30th October consistent with braise and fry high-fat foods that known to generate abundant aerosols (Xu et al., 2024). Therefore, although both AirGradientONE and AtmoCube sensors correctly identified the timing of each





emission episode, AirGradientONE consistently reported higher absolute concentrations, particularly for the most intense combustion plumes than those of AtmoCube sensors.

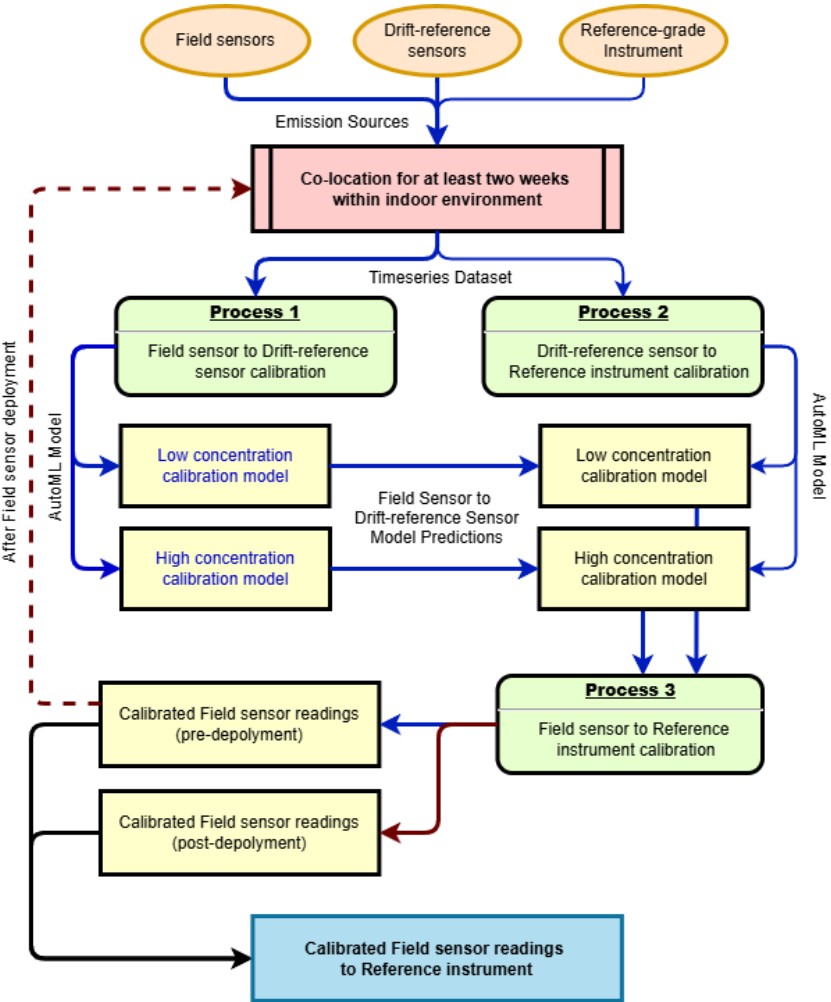

**Figure 2: Flowchart of the indoor air quality sensor calibration strategy.**

The inter-type relationship is summarised in Fig. S2, showing the averaged drift-reference $PM_{2.5}$ measurements from AirGradientONE and AtmoCube. At concentrations below ~50 µg m$^{-3}$ (hereafter denotes as "below-50") (Fig. S2(a)), AirGradientONE readings lay predominately above the 1:1 reference line, showing a positive bias relative to AtmoCube sensors. Once concentrations exceeded ~50 µg m$^{-3}$ (denotes as "above-50") (Fig. S2(b)), this coherence vanished and the paired data became more scattered, indicating that the two sensor types diverge progressively with increasing particle load.

Calibration that reconciles these type- (brand) specific sensitivities is therefore essential for any application that requires accurate absolute $PM_{2.5}$ values.



Sensor-measured environmental parameters exhibited similar systematic offsets (Fig. S3 for temperature and Fig. S4 for RH). Throughout the calibration, AirGradientONE temperatures were 1.2–1.8°C higher than those from AtmoCube (Fig. S3(a) and
S3(b)), where paired data cluster above the identity line (slope=1.01, $R^2$=0.94). AirGradientONE measured 4–7 % lower than AtmoCube sensors for RH maxima, whereas at minima AirGradientONE read 3–5 % higher, as in Fig. S4(a) and S4(b). Intra-type variability reached ~2°C for AirGradientONE sensors but was ≤1.5°C for AtmoCube sensors, and both types recorded the same diurnal trend (Fig. S3(c) and S3(d)). RH measurements ranged from 47% to 89% (Fig. S4(c) for AirGradientONE and 4(d) for AtmoCube). AirGradientONE sensors exhibited tighter clustering (intra-type variability ≤5%) than AtmoCube
(≤10%), but they showed a systematic pattern.

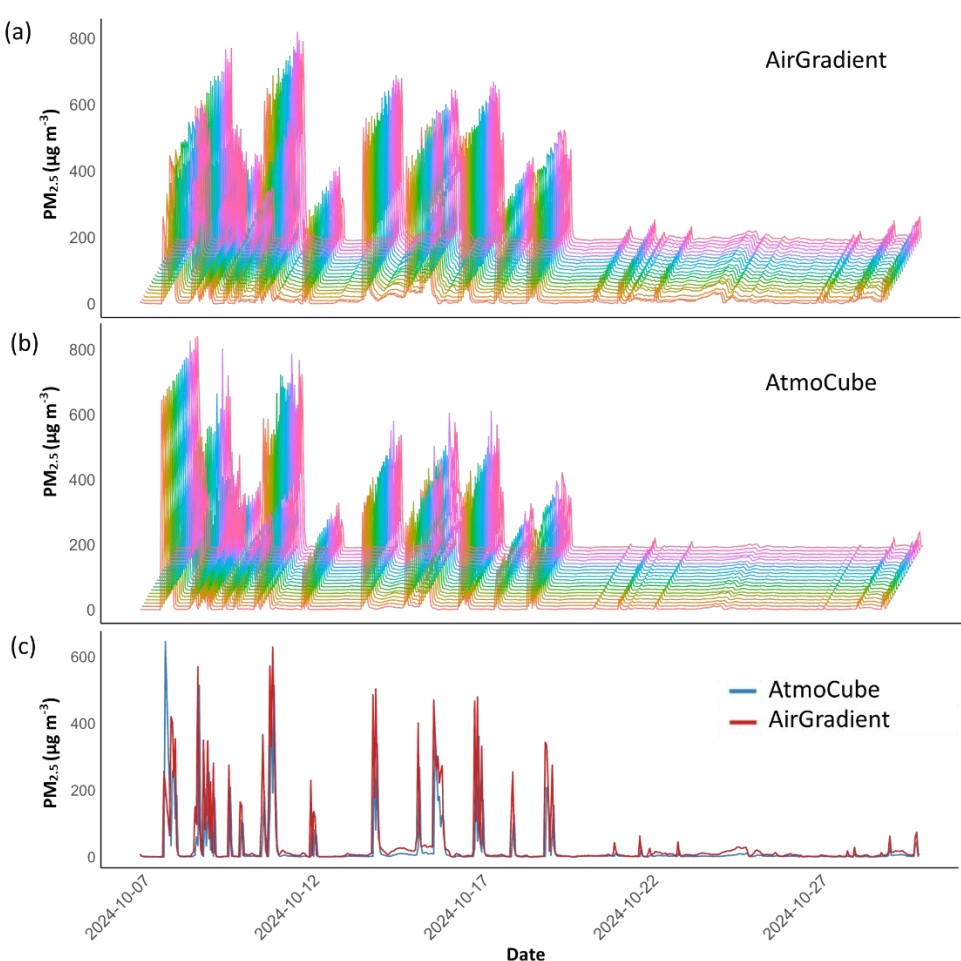

**Figure 3: Timeseries of (a) PM$_{2.5}$ from AirGradient (AirGradientONE) sensors, (b) PM$_{2.5}$ from AtmoCube (AtmoCube) sensors, and (c) Averaged concentration from drift-reference sensors.**




### 3.2 Raw readings from drift–reference sensors vs. Fidas 200 measurements

Figures 4(a) and 4(c) shows scatter plots of raw and calibrated averaged $PM_{2.5}$ concentrations from AirGradientONE and AtmoCube drift-reference sensors against the Fidas 200 measurements in the below-50 regime, representative of relatively low air pollution. Before calibration, both AirGradientONE and AtmoCube sensors exhibited moderate linear correlations with
the Fidas 200, with $R^2$ values of 0.65 for the AirGradientONE and 0.57 for the AtmoCube, respectively (Table 1). Although both sensor types clustered close to the 1:1 reference line, their slopes reveal systematic biases. AirGradientONE readings lay predominately above the line with a regression slope of 1.57, producing an average 20% overestimation relative to the Fidas 200, while AtmoCube readings fell below with a slope of 0.64, corresponding to a 55.6% underestimation. Extending the analysis to the above-50 regime (Figs. 4(b) and 4(d)) highlights further divergence. Here, AirGradientONE sensors had a
stronger correlation with the reference ($R^2$=0.78), but its slope decreased to 0.82, reflecting a slight 3.1% underestimation during high pollution episodes. In contrast, AtmoCube sensors had a lower slope of 0.50 and an $R^2$ of 0.64, showing a substantial 38.8% underestimation. Therefore, both types of sensor experience signal compression at higher particle loads, yet the magnitude of this non-linearity is sensor specific.

RH can significantly influence the measurement accuracy of particles from indoor air quality sensors (Fig. S5). For
AirGradientONE (Fig. S5(a) and S5(b)), $PM_{2.5}$ readings above the 1:1 reference line at low concentrations consistently associated with periods of high RH, implying that hygroscopic growth of particles at high humidity is a primary driver of AirGradientONE's low end overestimation (Liang, 2021). Conversely, AtmoCube showed no systematic RH pattern (Fig. S5(c) and S5(d)); its scatter remained broadly uniform across the humidity spectrum, indicating lower RH sensitivity. This disparity may reflect differences in internal RH-compensation algorithms implemented by each manufacturer.

**Table 1: Statistical performance of raw and calibrated AirGradientONE and AtmoCube drift–reference sensors relative to the Fidas 200 measurements for $PM_{2.5}$, stratified by concentration regime (below-50, above-50) and for the combined dataset.**

| Sensor | Subset | Stage | n (sample size) | $R^2$ | RMSE | MAE | MBE | IOA |
|---|---|---|---|---|---|---|---|---|
| AirGradientONE | Below 50 µg m⁻³ | Raw | 483 | 0.65 | 6.4 | 3.7 | -1.8 | 0.49 |
| | | Calibrated | 483 | 0.69 | 3.8 | 1.5 | 0.1 | 0.80 |
| | Above 50 µg m⁻³ | Raw | 64 | 0.78 | 91.3 | 69.6 | 40.9 | 0.80 |
| | | Calibrated | 64 | 0.92 | 59 | 44.6 | -5.4 | 0.87 |
| | All concentration range | Raw | 547 | 0.95 | 31.8 | 11.4 | 3.2 | 0.90 |
| | | Calibrated | 547 | 0.97 | 20.5 | 6.5 | -0.6 | 0.94 |
| AtmoCube | Below 50 µg m⁻³ | Raw | 499 | 0.57 | 12.4 | 5.1 | 0.04 | 0.63 |
| | | Calibrated | 499 | 0.80 | 7.4 | 2.8 | 0.03 | 0.79 |
| | Above 50 µg m⁻³ | Raw | 48 | 0.64 | 182.7 | 160.5 | 150.8 | 0.48 |
| | | Calibrated | 48 | 0.76 | 91.1 | 72.3 | 2.4 | 0.76 |





| | | | | | | | |
|---|---|---|---|---|---|---|---|
| All concentration range | Raw | 547 | 0.90 | 55.4 | 18.7 | 17.7 | 0.84 |
| | Calibrated | 547 | 0.94 | 27.9 | 8.9 | 0.3 | 0.92 |

## 3.3 Calibrated readings from drift-reference sensor vs. Fidas 200 measurements

In the below-50 regime, calibrated AirGradientONE drift-reference readings show slight stronger correlation with Fidas 200
measurements ($R^2$=0.69) compared to their raw values (Fig. 4(a)), and errors are relatively small (RMSE=3.8 µg m$^{-3}$,
MAE=1.5 µg m$^{-3}$) as in Table 1. The residuals present negligible systematic bias (MBE=0.1 µg m$^{-3}$), indicating great
improvements from systematic overestimation under low PM$_{2.5}$ concentration before calibration. After calibration, the sensor
performance meets the recommended criteria of $R^2 \geq 0.70$ and RMSE $\leq 7$ µg m$^{-3}$ (Zamora et al., 2022). At above-50
concentrations (Fig. 4(b)), the improvement in the performance of calibrated AirGradientONE sensors was even more
significant, with $R^2$ and IOA achieving about 0.92 and 0.87, respectively. Absolute errors increase (RMSE = 59 µg m$^{-3}$, MAE
= 44.6 µg m$^{-3}$) as expected but remain proportionally reasonable (e.g., ~10 at 600 µg m$^{-3}$). A slight negative bias (MBE = -
5.4 µg m$^{-3}$) indicating a small underestimation tendency at extreme high concentrations, but high IOA value (0.87) show
accurate tracking of both timing and magnitude.

Figure S6 show the impact of RH on calibrated readings of AirGradientONE sensors for the below-50 (Fig. S6(a)) and the
above-50 (Fig. S6(b)) concentration regimes, respectively. Across both concentration ranges the residuals show no systematic
humidity bias, indicating that the AutoML model (using RH and temperature as covariates) mitigated hygroscopic growth
influences that typically inflate optical counts above 70–80% RH (Ko et al., 2024). The small scatter evident at extreme high
RH levels likely reflects limited training data but does not compromise agreement with the reference, corroborating reports
that RH-aware calibration can suppress sensor error by around 20% (Liang, 2021).

Calibration likewise improved AtmoCube agreement with the Fidas 200 across the full concentration range (Figs. 4(c)
and 4(d)). Overall AtmoCube sensors achieved $R^2$=0.94 and IOA=0.92 (Table 1). In the below-50 clean air conditions, the
calibrated AtmoCube sensors have $R^2$=0.80, and such slightly lower correlation relative to those of high pollution levels is
expected as sensor signals approach the noise floor at very low pollution levels (Johnson et al., 2018). RMSE (7.4 µg m$^{-3}$) and
MAE (2.8 µg m$^{-3}$) are relatively small, and the mean bias is negligible, indicating that the calibration mitigates the pronounced
low-end under-reading observed pre-calibration. At high PM$_{2.5}$ levels, calibrated AtmoCube sensors still show good agreement
with Fidas 200 as data points distribute along the 1:1 line but with slightly reduced $R^2$ (0.76). A possible explanation is that at
very high particle loading the sensor's optical detector response starts to become non-linear or approaches a saturation point
(Kelly et al., 2017), introducing larger random errors. The residual bias is minor (MBE=2.4 µg m$^{-3}$), indicating a small over-
read under very high pollution. Figure S6(c–d) shows that, after calibration, AtmoCube residuals remain almost flat across the
full RH ranges in both low and high concentration regimes. Even during episodes exceeding 80 % RH, no coherent over- or
under-reading trend was found, indicating that the calibration has effectively reduced humidity interference.




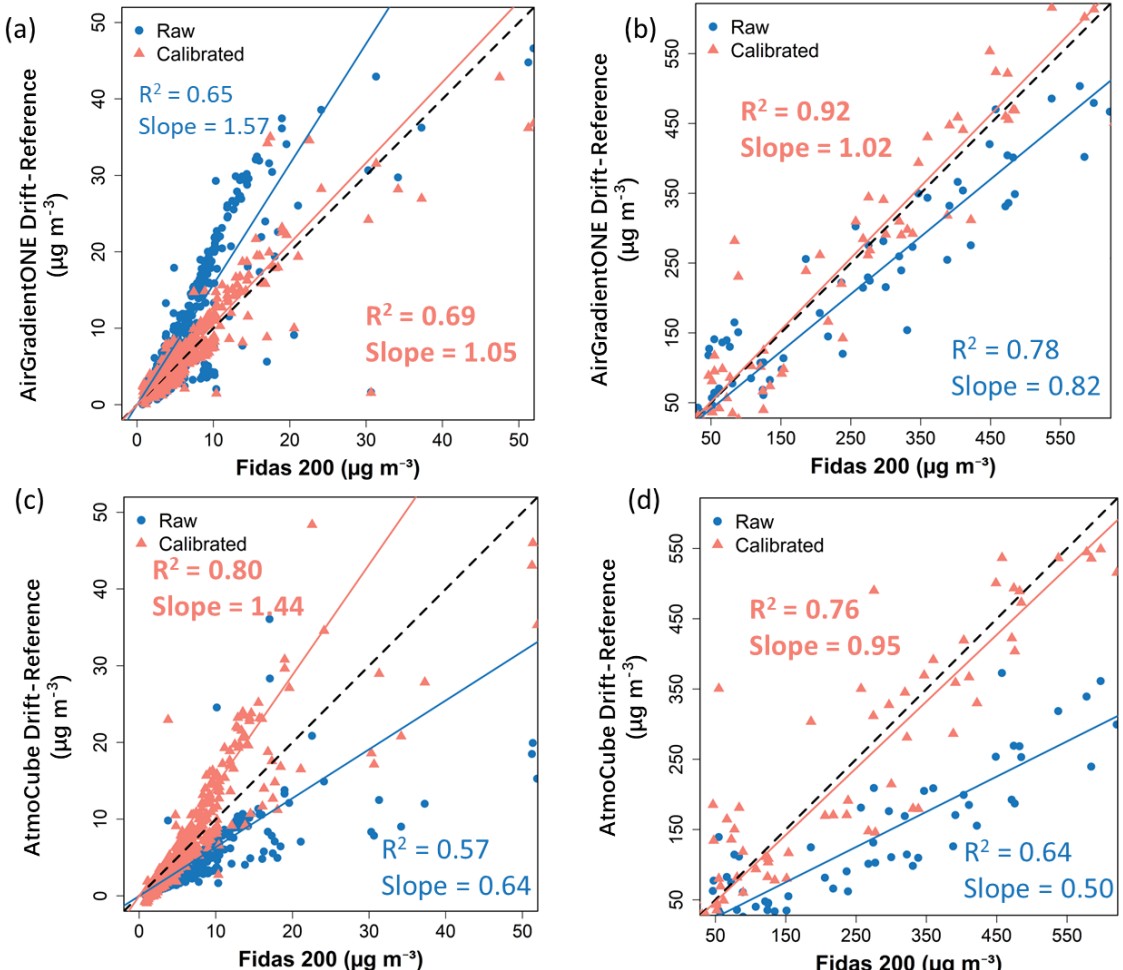

**Figure 4: Raw and calibrated PM2.5 of drift-reference sensors compared with the Fidas 200 measurements, (a) AirGradientONE sensors within below-50 regime; (b) AirGradientONE sensors within above-50 regime; (c) AtmoCube sensors within below-50 regime; (d) AtmoCube sensors within above-50 regime.**

**3.4 Calibrated readings from field sensors vs. Fidas 200 measurements**

The multi-stage calibration strategy effectively improved the performance of field sensors against the reference-grade instrument Fidas 200 (Fig. 5 and Table 2). Within the below-50 regime, AirGradientONE sensors showed a RMSE of 4 µg m⁻³ and MAE of 1.70 µg m⁻³, and their correlation $R^2$ increased from 0.45 to 0.64. By contrast, AtmoCube sensors achieved a stronger linear match ($R^2$=0.80) despite relatively higher residual scatter (RMSE=7.5 µg m⁻³) (Fig. 5(c)), consistent with their finer baseline sensitivity to subtle particulate variations. Performance at above-50 concentration regime indicated that both types of indoor air quality sensor synchronised well with the timing of pollution events while their error signatures differed.





AirGradientONE sensors showed moderate overestimation (MBE=3.9 µg m⁻³, RMSE=67.1 µg m⁻³), while AtmoCube sensors

displayed similar systematic bias (MBE=3.7 µg m⁻³) but higher variability (RMSE=91.5 µg m⁻³). These differences may arise from different sensor components, for example, AtmoCube units employed shorter optical path length and proprietary firmware averaging while AirGradientONE sensors used longer path and raw count reporting of the Plantower PMS5003. Importantly, our calibration strategy reconciled hardware-driven disparities between sensor types. Both types of sensors agreed well with Fidas 200 measurements after calibration, with IOA increasing from 0.90 to 0.94 for AirGradientONE and from 0.84

to 0.92 for AtmoCube sensors.

To evaluate the multi-step calibration strategy itself rather than the choice of models, we compared AutoML models with multivariate regressions (Fig. S7). Figure S7(a) shows that AutoML models produced better performance statistics, showing enhanced predictive accuracy and reliability, particularly when evaluating error distribution across different $PM_{2.5}$ concentration regimes. MAE (Fig. S7(b)) reduced by 15-40% across different concentration ranges, with the largest

improvement happened in the 25–50 µg m⁻³. Such improvements could be due to the ability of AutoML to incorporate interaction terms (RH, temperature) that influence the sensor light-scattering response (Liang, 2021).

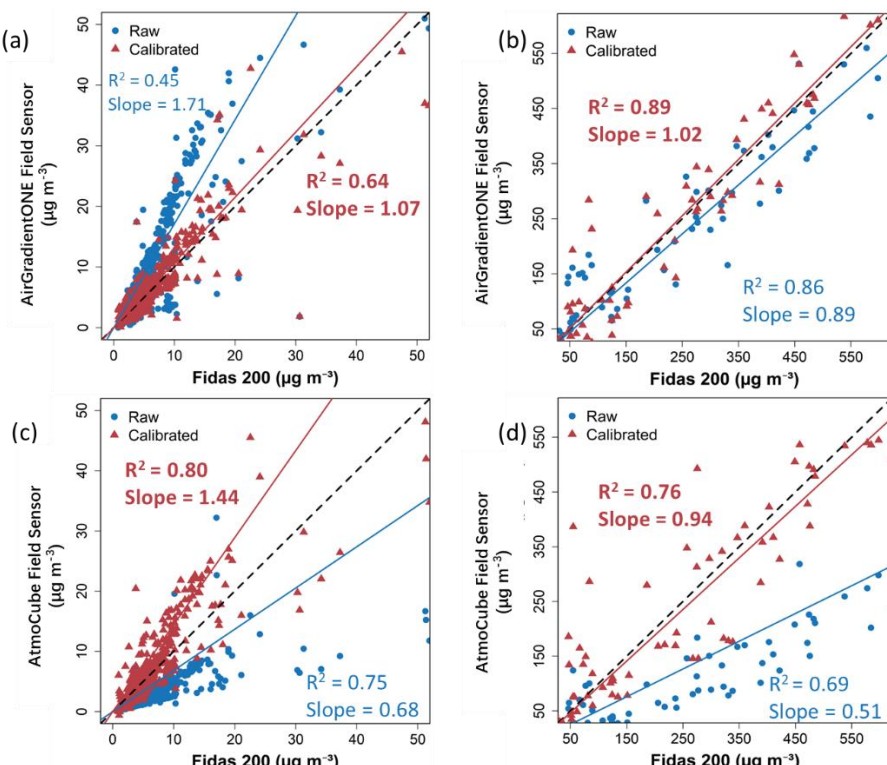

**Figure 5: Raw and calibrated $PM_{2.5}$ of field sensors compared with the Fidas 200 measurements, (a) AirGradientONE**

**sensors within below-50 regime; (b) AirGradientONE sensors within above-50 regime; (c) AtmoCube sensors within below-50 regime; (d) AtmoCube sensors within above-50 regime.**





**Table 2: Statistical performance of raw and calibrated AirGradientONE and AtmoCube field sensors relative to the Fidas 200 measurements for PM$_{2.5}$, stratified by concentration regime (below-50, above-50) and for the combined dataset.**

| Sensor | Subset | Stage | n | R$^2$ | RMSE | MAE | MBE | IOA |
|---|---|---|---|---|---|---|---|---|
| AirGradientONE | Below 50 µg m$^{-3}$ | Raw | 483 | 0.45 | 7.3 | 4.3 | 2.3 | 0.41 |
| | | Calibrated | 483 | 0.64 | 4 | 1.7 | 0.1 | 0.77 |
| | Above 50 µg m$^{-3}$ | Raw | 64 | 0.86 | 83.1 | 62.2 | -22.2 | 0.82 |
| | | Calibrated | 64 | 0.89 | 67.1 | 48.7 | 3.9 | 0.86 |
| | All range | Raw | 547 | 0.94 | 29.2 | 11 | -0.6 | 0.90 |
| | | Calibrated | 547 | 0.96 | 23.3 | 7.2 | 0.5 | 0.94 |
| AtmoCube | Below 50 µg m$^{-3}$ | Raw | 499 | 0.75 | 12.4 | 4.9 | -4.6 | 0.64 |
| | | Calibrated | 499 | 0.80 | 7.5 | 3.6 | 0.2 | 0.77 |
| | Above 50 µg m$^{-3}$ | Raw | 48 | 0.69 | 180.3 | 158.2 | -147 | 0.48 |
| | | Calibrated | 48 | 0.76 | 91.5 | 72.6 | 3.7 | 0.74 |
| | All concentration range | Raw | 547 | 0.88 | 54.7 | 18.3 | -17.1 | 0.84 |
| | | Calibrated | 547 | 0.94 | 28.1 | 9.6 | 0.5 | 0.92 |

## 3.5 Limitations and implications

Our framework significantly improved the low-cost sensors performance under different concentrations. But there are still some limitations, and further research is needed on the generalizability of the model and calibration strategies. First, the training data were collected in a single experimental container under temperate-climate humidity (with RH between 45–85%) and may not capture sensor behaviour in very moist interiors. Second, the present study did not capture every indoor emission source, particularly those with moderate emission levels. We do not know whether the sensors will be sensitive to particle types (e.g., particles from different sources). Third, evaluating sensor drift demands the months-to-years timescales of real deployments and was not evaluated. Future work should gather data from warmer, high-humidity homes to capture sensor behaviour at elevated RH conditions, consider additional moderate emission sources such as off-gassing materials, and run multi-year field trials to quantify drift and test automated recalibration. These steps will increase the robustness and evaluate long-term accuracy of the calibration strategy. However, the thresholds delineating "low" and "high" categories are derived from empirical observations within the analysed dataset. Accordingly, researchers are encouraged to initially assess their own data and adapt this strategy as necessary to ensure its applicability.

The implications of our findings are significant for atmospheric science and indoor air quality management, especially in the context of the growing use of low-cost sensors for exposure assessment and public health applications. By showing that inexpensive sensors can be calibrated to yield high-quality data indoors, this study helps bridge the important gap between indoor and outdoor air pollution monitoring. Furthermore, the application of AutoML in sensor calibration showcases the value of advanced data-driven techniques in atmospheric measurements. AutoML could be used to periodically re-calibrate hundreds of sensors automatically as new reference data become available, maintaining network accuracy with minimal human intervention. This is particularly relevant for community science projects or indoor air quality campaigns where resources for





manual calibration are limited. By improving the reliability of indoor air measurements, the study contributes to a future where continuous indoor air quality monitoring is feasible on a large scale, driving better-informed strategies to safeguard public health in the spaces where people live and work.

## 4 Summary

In this work, we introduced an automated machine learning (AutoML) calibration framework for enhancing the performance
of low-cost indoor air quality sensors. The AutoML-calibrated sensors met or exceeded study objectives by significantly improving measurement accuracy for fine particles (PM$_{2.5}$) across all concentration regimes. The multi-stage calibration workflow achieved tight agreement with reference measurements (from Fidas 200), evidenced by substantial increases in coefficient of determination ($R^2$) and reductions in error metrics. In the low-concentration regime (below 50 µg m$^{-3}$), R2 improved from moderate values (~0.6 pre-calibration) to approximately 0.85 post-calibration, with root-mean-square error
(RMSE) dropping by roughly half (e.g., from ~5 to ~3 µg m$^{-3}$). At higher concentrations (above 50 µg m$^{-3}$), gains were even more pronounced, with $R^2$ approaching or exceeding 0.90 (near reference-grade performance) and RMSE falling from tens of µg m$^{-3}$ to single digits. Similarly, mean absolute error (MAE) declined markedly, and mean bias error (MBE) was effectively eliminated, shifting from significant systematic biases (e.g., 5–10 µg m$^{-3}$ over- or underestimation) to nearly zero. These results show that the calibrated sensors reliably resolve indoor particulate levels at background concentrations and
during elevated pollution events, closely tracking the reference instrument across the full range. These findings confirm that our multistage calibration effectively eliminated sensor bias under varied indoor conditions and emission sources. The initial stage corrected baseline drift. Subsequent stages used AutoML to address scatter caused by relative humidity and nonlinear responses at high particle concentrations. These factors are often overlooked in simpler methods. AutoML efficiently selected the best models for each phase, removed the need for manual tuning, and revealed subtle patterns in the data. By
integrating AutoML into a structured multistage process, we achieved robust bias correction across scenarios, yielding accurate, precise measurements well-suited for indoor air quality monitoring.






**Author Contributions**

**Juncheng Qian:** Writing – original draft, Writing – review & editing, Visualization, Methodology, Investigation, Formal analysis, Data curation, Conceptualization. **Thomas Wynn:** Writing – review & editing. **Bowen Liu:** Writing – review & editing, Supervision. **Yuli Shan:** Writing – review & editing, Supervision. **Suzanne E. Bartington:** Writing – review & editing. **Francis D. Pope:** Writing – review & editing. **Yuqing Dai:** Writing – review & editing, Visualization, Methodology, Investigation, Formal analysis, Data curation, Conceptualization, Supervision. **Zongbo Shi:** Conceptualization, Interpretation, Visualization, Writing – review & editing, Supervision.

**Acknowledgement:**

We would like to thank Joseph Day and Nana Wei, and the wider WM-NetZero research team for their support, including comments on the manuscript. This work was supported by Wellcome Trust grant number: 227150_Z_23_Z, and UKRI-MRC grant number: MR/Z506680/1.

**Declaration of competing interest**

Some authors are members of the editorial board of journal Atmospheric Measurement Techniques.

**Data availability**

Data is available at https://github.com/DandE9996/sensor_calibration



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
