# Peer review of "Enhancing Accuracy of Indoor Air Quality Sensors via Automated Machine Learning Calibration"

_EGUsphere, 2025_

## Author Response (AR1)

**Response to reviewers**

**Title: "Enhancing Accuracy of Indoor Air Quality Sensors via Automated Machine Learning Calibration"**

General Response: We would like to thank the editor and reviewers for the positive feedbacks and constructive comments. Below, we've provided a point-by-point response to all comments to clarify revisions and improvements in the manuscript.

**Reviewer #1:**

**Comment 1. Ln 171: As far as I understand, there are eight regression algorithms. Better to mention the exact number of algorithms you used instead of 'multiple'.**

Response:

We thank the reviewer. We have revised the Methods to report the exact algorithms used and to clarify wording. Specifically, we now state that AutoML evaluated 30 machine learning algorithms (Table S1).

*Revised sentences:*

*We employed an AutoML framework to develop and select calibration models for the indoor air quality sensors. The AutoML approach generate a variety of (i.e., 30 in this study) candidate models and optimised their hyperparameters. Then the AutoML algorithm would identify a model that best maps the sensor outputs to the reference concentrations by ranking the cross-validation root mean squared error (RMSE). In our implementation, the input features to each model included the sensor's raw readings, indoor temperature, and RH, while the target output was the PM concentration measured by Fidas 200.*

**Comment 2. Ln 175: How did you allocate the 80% and 20% dataset? Randomly, chronologically, or other methods? Did you also use cross validation in the step?**

Response:

Data splitting was random following previous studies. Within each concentration regime (<50 and ≥50 $\mu g\ m^{-3}$), we used H2O's splitFrame with a fixed seed (1014) to allocate 80% of the rows to training and 20% to a held-out test set. During AutoML, we used k-fold cross-validation (5-fold) on the training portion for model selection (sorted by RMSE). The held-out 20% test set was never used for training or tuning; we report both cross-validated training metrics and external test metrics (see Table S1). This choice ensured both train/test and cross-validation folds contained comparable concentration distributions while avoiding temporal leakage, as the experiment container was well-mixed and emission episodes were interleaved.

We revised the manuscript by adding the above paragraph to reflect these points.

**Comment 3. Ln 223: From Figure S1, I can't really see a clear threshold of 50 $\mu g\ m^{-3}$. To me, it looks more like 25 $\mu g\ m^{-3}$ is the threshold. It's ok to use 50 $\mu g\ m^{-3}$, but it's better to show more clearly why it is chosen.**

Response:

Thank you for the helpful observation. We chose 50 $\mu g\ m^{-3}$ because the scatter shows two regimes relative to the 1:1 line. At or below 50 $\mu g\ m^{-3}$ the point cloud is tight and lies mostly above the 1:1 line. This indicates a positive sensor bias at low concentrations. Above 50 $\mu g\ m^{-3}$ the points shift below the 1:1 line and the fitted trend is flatter

than the 1:1 reference. This is consistent with signal compression at higher particle loads. The observations are not evenly distributed. Most data lie below about 25 $\mu g\ m^{-3}$ and only a small number of points fall between 25 and 100 $\mu g\ m^{-3}$. A split at 50 $\mu g\ m^{-3}$ therefore separates the two behaviours cleanly and avoids further fragmenting ranges that are already sparse. We have clarified this in the manuscript so that the rationale for the chosen split is explicit.

*Revised sentences:*

*Our exploratory analysis (Fig. S1) revealed a clear threshold at 50 $\mu g\ m^{-3}$ where the sensor bias flips. We chose this value because the scatter plot of sensor versus reference measurements shows two distinct regimes relative to the 1:1 line. At or below 50 $\mu g\ m^{-3}$, the data cloud is tight and lies mostly above the 1:1 line, which indicates a positive sensor bias (overestimation) at low concentrations. Conversely, above 50 $\mu g\ m^{-3}$ the cloud shifts below the 1:1 line, and the fitted trend becomes flatter than the 1:1 reference, a pattern consistent with signal compression and underestimation at higher particle loads. This split is further justified by the data distribution; most data lie below about 25 $\mu g\ m^{-3}$, with only a small number of points between 25 and 100 $\mu g\ m^{-3}$. A split at 50 $\mu g\ m^{-3}$ produces two interpretable regimes that align with the observed change in bias, keeps the rare high-concentration events together, and avoids slicing the dense background data into very small groups, which would reduce model stability. Therefore, we applied a stratified calibration strategy, training separate AutoML models for the low (<50 $\mu g\ m^{-3}$) and high (50–600 $\mu g\ m^{-3}$) regimes in both the field-to-drift (f2d) and drift-to-reference (d2r) stages. This allows us to tailor the calibration to the specific bias profile of each regime and thereby minimises systematic error across the sensor's full operating range.*

**Comment 4. Figure 2: It is a good flow chart, but the arrows with different colors are a bit confusing. What do brown and blue arrows represent respectively?**

Thank you for raising this point. We agree that the color encoding should have been explicitly defined in the manuscript. We have added clear explanations in Figure 2 caption and in the caption so that the meaning of each arrow is unambiguous.

The blue arrows represent the primary data flow used to train and apply the AutoML calibration models across Processes 1 to 3. This path links the field sensors, the drift-reference sensors, and the reference instrument during both training and prediction, and it yields calibrated field-sensor readings relative to the reference instrument. The brown dashed arrow represents the post-deployment recalibration path. After the field sensors are retrieved from the observation, we rebuild the field-to-drift and drift-to-reference models using the post-deployment dataset to correct for sensor drift and then route the updated predictions through Process 3 to obtain post-deployment calibrated readings referenced to the same instrument. The color scheme therefore distinguishes workflow timing rather than sensor type or data source, with blue indicating the main training and prediction workflow and brown dashed indicating the optional recalibration that is activated after deployment.

To make this explicit without modifying the flowchart, we inserted the following text in Figure 2 caption: "We use a fixed color scheme to distinguish the two stages of the workflow. Blue arrows and lines represent the main training and prediction path that spans Processes 1–3. The brown arrows and lines represent the post-deployment recalibration path, which is executed after sensor retrieval to correct drift using the post-deployment dataset. The resulting predictions are passed through Process 3 to obtain calibrated readings mapped to the reference instrument."

**Comment 5. Figure 3 a/b: Are these two subplots 3-dimensional? I see that all the lines for timeseries are not really aligned vertically. It can be misleading as they can mean that they are not in the same timestamp. I understand that in this way you can illustrate the difference between lines more clearly, but at least you need to draw a z-axis to show that there is a third dimension.**

Response:

Thank you for flagging this. In the revision we converted panels 3a and 3b into explicit 3D time series plots by adding and labelling a z axis. The x axis shows time, the y axis shows $PM_{2.5}$ concentration in $\mu g\ m^{-3}$, and the z axis indexes the sensors with tick labels corresponding to sensor numbers. All series share the same time coordinate, so timestamps are aligned vertically, and the apparent separation is now clearly along the sensor dimension rather than time. These clarifications improve readability and remove the possibility of misinterpretation, while leaving the analysis and conclusions unchanged. Please see the revised figure below.

[Figure]

**Comment 6. Table 1: The problem with using 50 µg m$^{-3}$ as threshold might be that the subset of 'above 50 µg m$^{-3}$' has too few data points compared to the 'below 50 µg m$^{-3}$'. Have you considered using another value as a threshold (similar to question 2)?**

Response:

We recognize that there are fewer data points above 50 µg m$^{-3}$. We chose 50 µg m$^{-3}$ as the threshold because there is a clear, empirically observed transition between two distinct response regimes of the sensors in comparison with the reference instruments. As shown in our exploratory plots, data at or below this value forms a tight cluster predominantly above the 1:1 line, indicating a consistent low-end positive bias. In contrast, above 50 µg m$^{-3}$, the data points rotate below the 1:1 line and the slope flattens. Choosing the threshold at this transition point allows us to model a real change in the sensor's behavior, a rationale we now explicitly state in the Methods.

For data above-50 µg m$^{-3}$, although relatively small, it is sufficiently informative to train a stable model. The performance of the calibrated models validates this approach, as they retain strong agreement with the reference instrument. In this high-concentration regime, AirGradient achieved an $R^2$=0.92 and IOA=0.87, while AtmoCube achieved an $R^2$=0.76 with minor residual bias. The fit is neither noisy nor over-tuned. Further evidence comes from independent train-test statistics, which show that the models generalize well, achieving a test $R^2$ of 0.78–0.79 and confirming that the sample is large enough to learn a robust mapping specific to that regime.

We indeed considered adjusting the threshold, but the highly skewed nature of the data distribution makes alternatives ineffective. Most observations fall below 25 µg m$^{-3}$, with very few events between 25 and 100 µg m$^{-3}$. Lowering the threshold would contaminate the high-range model with low-bias data points, blurring the very change in slope we aim to capture. Conversely, raising the threshold would further diminish the already scarce high-range data. Therefore, keeping the threshold at the empirically observed change point preserves interpretability and reduces model misspecification.

**Comment 7. Ln 176-195, Figure S7, and several instances in the paper: You separated two model subsets 'below 50 µg m$^{-3}$' and 'above 50 µg m$^{-3}$'. When you compared the results using MAE and RMSE, it's not easy to tell which one has a worse MAE as you have the two subsets with different reference values. It's the most obvious on Figure S7 that you tried to compare the errors in different concentration ranges using MAE. This comparison may not be the most appropriate as a smaller concentration apparently has a smaller MAE. What matters in this case is the error percentage. I suggest authors using mean absolute percentage error (MAPE) instead of MAE when comparing the two subsets. For RMSE, there is also an alternative NRMSE.**

Response:

Thank you for this helpful suggestion, which we have implemented in the revised manuscript. In addition to RMSE and MAE, we now report normalized RMSE (NRMSE) and symmetric mean absolute percentage error (sMAPE) for each concentration regime. NRMSE is defined as RMSE divided by the mean Fidas 200 concentration in the corresponding subset and expressed as a percentage, while sMAPE provides a symmetric percentage error that is more stable at very low concentrations than conventional MAPE. Figure S7 and the accompanying text (Lines 176–195) have been updated so that comparisons between the below 50 µg m$^{-3}$ and above 50 µg m$^{-3}$ regimes are based on NRMSE and sMAPE rather than MAE alone. This makes the relative error behavior across the two concentration ranges clearer and directly addresses the reviewer's concern about concentration-dependent absolute errors.

We retain RMSE and MAE to facilitate comparison with guideline values, manufacturer specifications, and previous

calibration studies, which almost always report absolute error metrics, and because they summarize complementary aspects of model performance (MAE reflecting the typical deviation and RMSE being more sensitive to occasional large errors). In the revision, NRMSE and sMAPE are therefore used to compare relative performance across concentration regimes, while MAE and RMSE are reported alongside to document the absolute magnitude and structure of the residuals.

**Comment 8. Table S1: What are the criteria of ranking the eight regression algorithms? For the best algorithm? For model subset of low conc., the one listed as rank 1 has a higher RMSE/MAE and lower R2 compared to the one listed as rank 2. For high conc., the one ranked as 5th has a better MAE/RMSE/R2 results than the one ranked top. Are you using some other criteria for the ranking? Please list them in the table as well. Also, please clarify what criteria you used in the main text as well.**

Response:

We recognize the need to be clear with the criteria. In the original Table S1, "Rank 1" referred to the H2O AutoML leaderboard order, which we configured to sort by cross-validated RMSE computed on the training set (k-fold cross-validation). In k-fold cross-validation, the training set is split into k parts; each model is trained on k-1 parts and evaluated on the remaining part, and the errors are averaged. RMSE ($\mu g \ m^{-3}$) penalizes large errors more than small ones, which is important for $PM_{2.5}$ where high concentrations matter.

In the same table, we also reported performance on an independent 20% test set that AutoML did not use for training or ranking. Because the leaderboard metric (cross-validated RMSE) and the external test metric (single-split RMSE on the test set) are computed on different data with different protocols, it is expected that the model at Leader Rank 1 may not have the lowest error on that test set. This is especially plausible in the high-concentration subset, where there are fewer data points and variance is higher, and when comparing metrics with different sensitivities (e.g., RMSE vs. MAE vs. $R^2$).

To remove ambiguity, we revised Table S1 to report two ranks for each subset. Leader Rank is the AutoML order by cross-validated RMSE on the training set and is our model-selection criterion. We use Leader Rank because (i) it preserves the independence of the 20% test set by not using it to choose the winner, avoiding optimistic bias; (ii) averaging errors across folds provides a more stable, lower-variance estimate than a single split; and (iii) RMSE penalizes large deviations, aligning with the scientific and regulatory importance of limiting large $PM_{2.5}$ errors. Alongside this, External Test Rank orders models by RMSE on the independent 20% test set and is included for transparency. In the Methods we now state explicitly that selection follows the leaderboard metric, whereas all performance reported in the Results refers to the test set (with splits created using a fixed random seed for reproducibility).

For transparency we also describe each column exactly as it appears in the revised Table S1. "Model_Subset" identifies whether the model belongs to the low concentration subset below 50 $\mu g \ m^{-3}$ or the high concentration subset at or above 50 $\mu g \ m^{-3}$. "Model_ID" is the H2O identifier that allows exact reproduction. "Algorithm" is the model family returned on retrieval. "Test_RMSE", "Test_MAE", and "Test_R2" report performance on the independent 20% hold-out set, with RMSE and MAE in $\mu g \ m^{-3}$ and $R^2$ dimensionless. "Train_RMSE", "Train_MAE", and "Train_R2" are training-frame summaries shown for context. All columns prefixed with "LB_" are copied directly from the AutoML leaderboard with extra columns enabled and therefore reflect the cross-validation view used to form the leaderboard. In this file they include "LB_rmse", "LB_mae", "LB_mean_residual_deviance", and "LB_ Rank", where "LB_Rank" is the raw leaderboard order that determines the top model for each subset.

**Comment 9. Could this framework also be used outdoors for ambient air concentration? Can this framework also work for aerosols of larger size and gas pollutants?**

Response:

Thank you for raising the question of generalizability. The framework is model agnostic and can be transferred to outdoor settings. In practice, the same three stage structure field to drift reference, drift reference to reference instrument, and the composite field to reference transfer can be implemented outdoors by collocating a subset of sensors with a regulatory or research-grade outdoor reference sensor and by using outdoor meteorology as covariates such as ambient temperature, relative humidity, and wind driven dispersion proxies. The concentration regime split is not fixed and should be learned from the outdoor dataset using the same data driven rationale that we applied indoors. However, if there is no clear difference in the trend in different concentration range, one ML model may be sufficient. The manuscript now states that regime boundaries are empirical and should be re-estimated for a new application, e.g., outdoors.

Yes, the workflow should also apply to other particulate matters such as $PM_1$, $PM_4$, and $PM_{10}$ as well as if a time aligned reference for the target parameter is available.

**Technical comments:**

**Ln 251: No need to use hyphens for 400-to-500 µg m-3**

**Ln 261: If you have used 50 µg m-3 as a threshold, then there is no need to use the symbol '~'.**

**Ln 311: What is ~10? It's very confusing what you referred to without a unit.**

Response:

Thank you for these careful technical comments. We have revised the text accordingly. At line 251, "between 400-to-500 µg m$^{-3}$" has been corrected to "between 400 and 500 µg m$^{-3}$". At line 261, we removed the tilde so that 50 µg m$^{-3}$ is now written without the approximation symbol, consistent with its use as a threshold. At line 311, we clarified that "~10" refers to an approximate 10% relative error at 600 µg m$^{-3}$ and now state the unit explicitly in the sentence.

**Reviewer #2:**

**Comment 1. Line 146–152: Here, the authors describe which two types of low-cost sensors were used and against which reference instrument they were compared. I am wondering whether the low-cost sensors have a similar detectable particle size range as the Palas Fidas 200. It could add an additional layer of clarity to mention the particle size range for the reference instrument and - if available - also for the low-cost sensors.**

Response:

Thank you for this suggestion.

We do agree. Yes, the detectable particle size range is similar. We have revised the manuscript to include the manufacturer-specified particle size ranges for all instruments. The reference instrument, the Palas Fidas 200, has a certified detection range of 0.18–18 μm. The low-cost sensors, the Plantower PMS5003 and the Sensirion SPS30, both have a specified particle size detection range of 0.3–10 μm.

While addressing this comment, we also took the opportunity to include the manufacturer-specified measurement uncertainties for all instruments to further enhance the technical comparison. The revised text now reflects both points.

Revised sentences:
*An aerosol spectrometer (i.e., Palas Fidas 200 (detectable particle size of 0.18-18 μg, ranges from 0 to 10,000 μg m$^{-3}$ with 9.7% uncertainty for PM$_{2.5}$ measurements)) was used as the reference-grade instrument for sensor performance evaluation and calibration. A total of 40 low-cost air quality sensors was deployed within the chamber, settled on a table at near the same height with Fidas 200 to minimize positional variability. Our air quality sensors consisted of two different types, including 20 units of AirGradient ONE (Model I-9PSL) and 20 units of AtmoCube. AirGradient ONE sensors measure PM$_{2.5}$ using a Plantower PMS5003 laser-scattering sensor (manufacturing specification: detectable particle size of 0.3-10 μm, with ±10 μg m$^{-3}$ at 0-100 μg m$^{-3}$ 10% at 100-500 μg m$^{-3}$), and temperature and RH through a Sensirion SHT40 sensor. AtmoCube sensors detect particulate matter using a Sensirion SPS30 laser-scattering sensor (manufacturing specification: detectable particle size of 0.3-10 μm, with ±5 μg m$^{-3}$ at 0-100 μg m$^{-3}$, ±10% at 100-1000 μg m$^{-3}$), temperature using a Sensirion STS35-DIS, and RH using a Sensirion SHTC3.*

**Comment 2. Line 154 – 155: Hopefully, the team got to enjoy the food afterward. Scientific dedication always deserves a good meal.**

Response:

Thank you for this kind remark. The cooking part did indeed come with some well-deserved meals for the team, and it was all conducted in accordance with our laboratory safety and hygiene procedures.

**Comment 3. Line 157: The phrase "natural indoor conditions" sounds somewhat contradictory, since indoor environments are by definition artificial. It might be clearer to use "realistic" or "typical".**

Response:

Yes, we fully agree that "natural indoor conditions" is by definition artificial and thus we have revised the wording accordingly.

In the revised manuscript, we now write "typical indoor conditions" at the corresponding location (former line 157), so the sentence reads:

*"Temperature and RH levels were allowed to exchange passively with the outdoor air with no mechanic ventilations or windows/door opening, mimicking indoor conditions where these parameters may fluctuate."*

This change clarifies our intended meaning and avoids the contradiction you highlighted.

**Comment 4. Line 223–228: I wonder how the threshold of 50 μg m$^{-3}$ was determined. The text mentions that the exploratory analysis revealed a bias flip at this concentration, but it could strengthen the explanation to briefly clarify why 50 μg m$^{-3}$ was selected as the cutoff.**

Response: Please also see response to Comment 3 by reviewer 1.

We appreciate this observation. We selected 50 μg m$^{-3}$ because the scatter plot clearly shows two distinct regimes relative to the 1:1 line. Below 50 μg m$^{-3}$, the points form a compact cluster mostly above the 1:1 line for AirGradient ONE sensors, indicating a positive sensor bias at low concentrations. Above 50 μg m$^{-3}$ the points shift below the 1:1 line and the fitted regression becomes shallower than the 1:1 reference, which is consistent with signal compression at higher particle loads. The data are also unevenly distributed. Most observations fall below about 25 μg m$^{-3}$ and only a small fraction lie between 25 and 100 μg m$^{-3}$. Using 50 μg m$^{-3}$ as the threshold therefore separates these two behaviors clearly while avoiding further subdivision of already sparse ranges. We have revised the manuscript to explain this rationale explicitly.

**Comment 5. Line 251: In the phrase "between 400-to-500 μg m$^{-3}$" the hyphen is unnecessary. It would read more clearly as "between 400 and 500 μg m$^{-3}$."**

Response:

Thank you for pointing this out. We have revised the wording accordingly and now write "between 400 and 500 μg m$^{-3}$" to improve clarity at Line 251.

**Comment 6. Figure 3: The two time series plots showing the raw data from multiple low-cost sensors are displayed as 3D graphics, which causes a slight misalignment between the sensor readings and the x–y axes, making the visualization somewhat confusing. I am not sure this is the most effective way to present the data, although it is not a major issue. I would generally recommend adding x-axis tick marks to indicate the days (with major ticks for the labeled days and minor ones for each individual day) and including grid lines especially in panels (a) and (b), which would help improve readability and reduce the visual confusion caused by the three-dimensional layout.**

Response:

We appreciate you drawing attention to the potential ambiguity in Figure 3. In the revised version we have converted panels 3a and 3b into explicit three-dimensional time series plots by adding and labelling a z axis. The x axis now represents time, the y axis shows PM$_{2.5}$ concentration in μg m$^{-3}$, and the z axis indexes the sensors, with tick labels corresponding to sensor numbers. All series share a common time coordinate, so timestamps are aligned vertically and the apparent separation is clearly along the sensor dimension rather than time. These changes improve readability and remove the scope for misinterpretation, while leaving the analysis and conclusions unchanged.

**Comment 7. Figures 4 and 5 appear to have different sizes and resolutions. Since both figures are quite similar, it would be visually more appealing and consistent to display them at the same size and resolution for better comparison and overall presentation quality.**

Response:

Thank you for this helpful suggestion.

We agree that having Figures 4 and 5 in the same size and resolution improves consistency and makes comparison easier. In the revised manuscript, we have adjusted both figures so that they are now displayed at the same size and resolution to enhance visual clarity and overall presentation quality. Please see the revised figure below.

[Figure]

**Comment 8. I really appreciate the thorough and transparent discussion of the limitations of the proposed method. The authors clearly acknowledge the constraints related to environmental conditions, emission sources, and long-term applicability. An analysis of long-term sensor drift would have provided valuable additional insights into the robustness of the calibration over extended periods. However, since this aspect is explicitly discussed as a limitation and identified as an important direction for future work, I find the current scope appropriate and well-justified for this study.**

Response:

We thank the reviewer for this positive and encouraging comment on our discussion of limitations. We fully agree that an explicit analysis of long-term sensor drift would provide valuable evidence on the robustness of the calibration over extended deployments. In future studies we plan multi-year field deployments to quantify drift and to test automated recalibration within the proposed framework.

---

## Author Response (AR2)

**Response to the editor**

**Title: "Enhancing Accuracy of Indoor Air Quality Sensors via Automated Machine Learning Calibration"**

General Response: We would like to thank the editor and reviewers for the positive feedbacks and constructive comments. Below, we've provided a point-by-point response to all comments to clarify revisions and improvements in the manuscript.

**Comment 1. Figures 3 and 4 have similar structure (two-color scatter plots) but look different. Specifically, the second one has some 'gray border lines' outside the figures and uses different colors. It also looks a bit lower resolution. If the authors could update Figure 4 to match the style of Figure 3 it would be an improvement.**

**Response:**

We thank the editor for the comment. We assume that the editor means Figure 4 and Figure 5 in the manuscript. We have adjusted the figures as suggested. Now they have the same color and resolution. They have identical styles now.

[Figure]

**Figure 4:** Raw and calibrated PM$_{2.5}$ of drift-reference sensors compared with the Fidas 200 measurements, (a) AirGradient ONE sensors within below-50 regime; (b) AirGradient ONE sensors within above-50 regime; (c) AtmoCube sensors within below-50 regime; (d) AtmoCube sensors within above-50 regime.

[Figure]

**Figure 5:** Raw and calibrated PM$_{2.5}$ of field sensors compared with the Fidas 200 measurements, (a) AirGradient ONE sensors within below-50 regime; (b) AirGradient ONE sensors within above-50 regime; (c) AtmoCube sensors within below-50 regime; (d) AtmoCube sensors within above-50 regime.

**Comment 2. Similarly to both original reviewers, I find Figure 2 hard to read. I suspect the authors wanted to show how well the different sensors of the same kind correlate with each other, but I find it hard to do this on the given plot (for example, try comparing the values of sensors #3 and #13 for a given timestamp). I think that both 3D plots can be replaced with 2D figures that have time as a the X axis, concentrations as the Y axis, one line per sensor type showing the mean concentration per timestamp, and a shaded area with the concentration standard deviation across sensors for each timestamp. This figure would show both the mean concentration and a sense of how the observations vary between different sensors of the same kind. Instead of standard deviation, it could be possible to use 25-75% interquartile distance or even min/max values, whatever is deemed to better represent the underlying data.**

**Response:**

We thank the editor for this comment. We have revised the figures accordingly. Each figure now shows a single line representing the mean PM$_{2.5}$ concentration for the sensor model, with a shaded band indicating the minimum to maximum range across sensors of the same model. A second panel below presents the corresponding standard deviation over time. The shaded bands are narrow because differences among sensors within the same model are very small, indicating strong within model agreement. We believe this presentation most clearly conveys the close consistency across sensors while retaining information on variability.